# Geometry-Dependent Elastic Flow Dynamics in Micropillar Arrays

**DOI:** 10.3390/mi15020268

**Published:** 2024-02-13

**Authors:** Oskar E. Ström, Jason P. Beech, Jonas O. Tegenfeldt

**Affiliations:** Division of Solid State Physics, Department of Physics and NanoLund, Lund University, P.O. Box 118, 22100 Lund, Sweden; oskar.strom@protonmail.com (O.E.S.); jason.beech@ftf.lth.se (J.P.B.)

**Keywords:** DNA waves, micropillar arrays, microfluidics, elastic turbulence, geometry, polarization, polymer solutions, porous media

## Abstract

Regular device-scale DNA waves for high DNA concentrations and flow velocities have been shown to emerge in quadratic micropillar arrays with potentially strong relevance for a wide range of microfluidic applications. Hexagonal arrays constitute another geometry that is especially relevant for the microfluidic pulsed-field separation of DNA. Here, we report on the differences at the micro and macroscopic scales between the resulting wave patterns for these two regular array geometries and one disordered array geometry. In contrast to the large-scale regular waves visible in the quadratic array, in the hexagonal arrays, waves occur in a device-scale disordered zig-zag pattern with fluctuations on a much smaller scale. We connect the large-scale pattern to the microscopic flow and observe flow synchronization that switches between two directions for both the quadratic and hexagonal arrays. We show the importance of order using the disordered array, where steady-state stationary and highly fluctuating flow states persist in seemingly random locations across the array. We compare the flow dynamics of the arrays to that in a device with sparsely distributed pillars. Here, we observe similar vortex shedding, which is clearly observable in the quadratic and disordered arrays. However, the shedding of these vortices couples only in the flow direction and not laterally as in the dense, ordered arrays. We believe that our findings will contribute to the understanding of elastic flow dynamics in pillar arrays, helping us elucidate the fundamental principles of non-Newtonian fluid flow in complex environments as well as supporting applications in engineering involving e.g., transport, sorting, and mixing of complex fluids.

## 1. Introduction

Viscoelastic effects are pervasive wherever liquids contain polymer molecules that are freely suspended in the solvent. These molecules contribute an elastic energy component to the overall behavior of the fluid, resulting in a range of dynamic behaviors, typically at high Deborah numbers, De, and low Reynolds numbers, Re. The Elasticity Number, El≡De/Re, describes the relative magnitude of elastic and viscous forces. In this situation, it is high, leading to flow effects such as non-inertial, elastic turbulence [1]. The altered flow properties have strong relevance for, for example, food processing, polymer manufacturing, soft matter engineering, water treatment [2], heat transfer [3], and enhanced oil recovery [4]. Adding polymers has been shown to result in increased mixing rates in small systems [5] where flows are otherwise laminar and mixing highly inefficient. Relevant biomedical examples of polymeric viscoelastic flows in porous media include biofilm formation [6] and pathogen clearance by mucus in the lungs [7,8]. Despite having a broad importance, elastic flow dynamics remain unexplored to a large extent.

To better understand the viscoelastic effects that lead to dynamic behaviors and that influence overall flow properties in porous media, simple model systems have been employed based on microfluidic channels containing single to entire arrays of obstacles in 1D, 2D, and 3D [9]. Although real-life pore throat diameters and pore bodies are typically found in the range 100 nm–10 μm and 200 nm–50 μm, respectively [10], models with small pore dimensions are typically complicated to fabricate and more difficult to study. Most work on elastic flow across porous media has, therefore, focused on models with larger dimensions. However, viscoelastic flow behavior on small scales is of great interest as the relative magnitude of the elastic forces is predicted to strongly depend on relevant device feature sizes (*L*) due to the scaling of the Elasticity Number, $El\propto L^{-2}$. In addition, with small device dimensions, the length scale of the polymers and the length scale of the device features overlap, something we identified as an important factor for the formation of waves [11]. We could, therefore, expect entirely different (elasticity-dominated) flow behavior in smaller models that are relevant to a wide range of applications.

There has been significant interest in understanding how the density and distribution of the pillars affect the elastic flow in 2D micropillar arrays. Early work on larger pillar arrays (*R* = 2.38 mm) showed that unsteady flow patterns emerged at a lower flow rate for a hexagonal array compared to a quadratic array [12]. More recently, a wide range of microscopic pillar array geometries has been investigated. Work on 1D arrays demonstrates the crucial importance of the spacing between pillars for the resulting flow pattern. Shi et al. found that smaller spacing in a 1D pillar array results in stronger flow fluctuations at similar Weissenberg numbers, Wi [13]. Browne et al. observed that for a large pore spacing, upstream vortices form, whereas for a small pore spacing, switching between two unstable flow states occurs [14]. Others have studied the flow dynamics of both ordered [11,15,16,17,18,19] and disordered [11,17,18,20,21] 2D pillar arrays. It is not fully understood how the geometry changes the dynamics. Kawale et al. found that the upstream instabilities in a hexagonal pattern were shaped as geometrical prisms, while in a quadratic pattern, they filled up the space between the pillars in the flow direction [16]. There have been reports of pillar geometry randomization to both delay [18] and promote [17] the onset of elastic turbulence. The latter work reported that at sufficiently high Wi, the flow becomes geometry-independent. Although much work has been conducted, the parameter space is vast, and many factors, such as flow in small-scaled arrays, remain unexplored.

We have recently reported the formation of regular macroscopic waves [11] in high-concentration DNA solutions flowing through quadratic arrays of cylindrical pillars in microfluidic channels and demonstrated that these waves interfere in the sorting of DNA in microfluidic devices based on deterministic lateral displacement (DLD) [22]. We also noticed that the formation of waves is highly sensitive to any disorder in the array as well as to the symmetry of the pillars [23]. We introduce arrays with hexagonal geometry here due to their relevance for pulsed-field separation of DNA [24,25]. A hexagonal array can also be interpreted as a DLD device with a periodicity of N=2. Similarly, one can interpret a quadratic array as a DLD with periodicity of N=∞, which is equivalent by symmetry to N=0 and N=1. To better understand the underlying mechanisms of the waves, we also introduce a sparse array where we can study the viscoelastic flow around individual pillars. We evaluate the flow of a concentrated λ DNA solution (400 μg/mL) in our arrays to study the resulting flow patterns and dynamics at both macro and micro scales. By tracking the dissolved polymer molecules themselves with fluorescent imaging, we can observe fluctuations in local sample concentration, and using polarization microscopy, we can observe their orientation and extension.

## 2. Materials and Methods

The devices are fabricated using standard microfluidics practices with replica molding [26,27] using similar designs and approaches as described in Ref. [11]. See details of the device design and fabrication in Appendix A. The pillar arrays are approximately 8 mm long, approximately 800 μm wide, and 10.9 ± 0.1 μm in depth. The pillars are approximately 7 μm in radius with a gap of approximately 4 μm between rows and columns of the array except for the sparse array. See Table 1 for details of all the dimensions for all arrays. The hexagonal array has half a unit cell in row shift for every other row, thereby making it slightly stretched in the lateral direction compared to a true hexagonal array. The array of sparsely distributed pillars was designed to be as open as possible without exceeding aspect ratios that might lead to the collapse of the PDMS device.

We work at a semi-dilute concentration (400 μg/mL, concentration/overlap concentration ratio, *C*/*C** ≈ 4, see calculation details in [11]) of λ-phage DNA in a solution of 5× Tris EDTA (TE) buffer. We have added a bisintercalating dye, YOYO-1, at a base-pair-to-dye ratio of 50:1 to visualize the DNA using an epi-fluorescent microscope. We flow the DNA through the array by applying a pressure gradient. See the details in the Appendix A: experimental setup including polarization microscopy in Appendix A, sample preparation in Appendix A, and image analysis and frequency analysis in Appendix A.

We define the Deborah number $De\equiv (u/L_{ch})\tau$, where *u* is the estimated mean flow velocity, $L_{ch}$ is the characteristic length scale (radius of pillars plus the gap between the pillars for quadratic, hexagonal and disordered arrays, and the gap, 100 μm, for the sparse array) and τ = 1.43 s is the measured relaxation time of the polymer (at 25 °C, see details of the measurement in [11]). Please note that τ and De are defined assuming the conditions to be ideal, the solution to be dilute, and in equilibrium. Also, the numbers vary over time and depend on the location in the array. Furthermore, the exact definitions of length scales and velocities may differ in the literature. The numbers presented here should, therefore, be considered approximate and can be difficult to compare to the numbers reported by other authors. The mean flow velocity, u=Q/A, is calculated from the flow rate, *Q*, and the cross-sectional area of the device is calculated based on the lateral pillar gap and the number of pillars in the lateral direction, $A=GhN_{x}$. Numbers are rounded off to two significant digits.

## 3. Results

Results are presented for flow across three array geometries: quadratic, hexagonal, and disordered. In addition, a sparse array was studied to illustrate the behavior around individual pillars. The resulting fluorescence intensities and their fluctuations are presented in real space, in time, as well as in the corresponding Fourier space. The orientation of the DNA is visualized using polarization microscopy. For a full understanding of the following discussion, we strongly encourage the reader to view Appendix A.

### 3.1. Array Geometry Dictates the Large-Scale Flow Patterns

At high flow velocities, quadratic arrays exhibit ordered device-scale waves with the peaks corresponding to higher concentration and extension with synchronized orientation (see Figure 1A). We thoroughly investigated these waves as described previously [11]. When every other row is shifted half an array pitch, resulting in the hexagonal lattice, different flow dynamics can be observed. The repetitive diamond pattern of waves with distinct wavelengths of different orientations, like those seen in the quadratic array, can no longer be observed. Instead, we observe chaotically distributed zig-zag patterns of varying size at high flow velocities (see Figure 1B). The zig-zag patterns undergo continuous reshaping, disintegration, and merging with other regions into new zig-zag patterns. As observed previously [11], when the pillar distribution is randomized, no large-scale fluctuations are visible even at the highest flow velocity Figure 1C. Some regions develop concentration variations that look wavelike, but they do not propagate very far through the array as they do in the quadratic or hexagonal arrays. To more easily observe the dynamic phenomena, see Figure 2 for kymographs and Appendix A.

In a sparse array where the pillars are separated by a large distance (100 μm), we observe that the flow around the pillars couples only in the flow direction and that large-scale patterns do not form (see discussion in Section 3.5 below).

### 3.2. Spatial and Temporal Frequency Analysis of Concentration Fluctuations

To characterize the periodic behavior of the waves, spatial and temporal Fast Fourier Transforms (FFT) are applied to the movie data. Specifically for the spatial Fourier transform, an FFT is applied to each frame, and the average of these FFTs is presented in Figure 3 at increasing flow velocities. The frequency spectra at high flow rates exhibit clear differences for the different array types. For the quadratic array, a clear directionality is evident, while for the hexagonal array, a broad spectrum is the result (Figure 3A,B) consistent with Figure 1. The disordered array (Figure 3C), however, does not exhibit any significant change in the frequency spectrum with the flow velocity. The temporal frequency of the concentration fluctuations is characterized to address two questions. First, the Fourier amplitudes of the waves are used to quantify the strength of the waves (see Figure 4). In this way, the start-up distance of the formation of the waves can be quantified. Second, the amplitude spectra exhibit a power law with different exponents for different cases (see Figure 5).

To quantify the establishment of the long-range synchronization of the flow patterns, the temporal frequency spectrum is characterized as a function of longitudinal position along the device channel (see Figure 4). As is clear also from Figure 1, the waves do not start immediately in the array. Instead, it seems that the flow needs to pass approximately 50 rows of pillars for waves to form in the quadratic and the hexagonal arrays.

The power law dependence of the fluctuations may give clues to their origins. The temporal Fourier transform is thus performed on a region of interest in the center of the device. The region of interest is a circular region with a diameter of 163 μm corresponding to approximately the width of the waves. The result is that when there are no waves, i.e., for the disordered array as well as for the low flow rates for the quadratic and hexagonal arrays, we have an exponent with values in the range −0.5 … −1.0. For higher flow rates where fluctuations develop, we have exponents ≈−1.5 for quadratic and ≈−2.0 for hexagonal. The details of the calculations are given in Appendix A, and the results are given in Figure 5.

An alternative way to analyze the data is to consider the relationship between the average pixel values and the corresponding standard deviations. Density heat maps for the different geometries at different flow velocities are given in Appendix A. As expected, the disordered array exhibits small fluctuations, whereas the quadratic and the hexagonal arrays have similar overall patterns with increasing standard deviations as the average pixel values increase. A closer look at the data reveals that for the hexagonal array, the fluctuations are stronger for a given average pixel value than for the quadratic array, as can be seen in Figure 1.

### 3.3. Persistent Flow Patterns Depend on Geometries

When averaging over a long time period (long in relation to the fluctuations in the system), clear differences between the array types emerge. Figure 6 shows 30 s time averages of fluorescence videographs. Please note that the brightness and contrast have been enhanced to show the differences. The quadratic array shows waves moving through the device without any local bias. Therefore, an average over sufficient time to allow several waves to pass will not produce any pattern. In contrast, the low-concentration depletion zones of the hexagonal and disordered arrays exhibit different distinct patterns. Although the waves in the hexagonal array seem to be more chaotic than those in the quadratic array, their movement is biased such that certain local areas of the array are avoided. The rows of the hexagonal array thus exhibit long-range order comprising a synchronized depletion zone with orientation either to the left or right of the general direction of the flow. The patterns in the disordered array are highly dependent on the local pillar arrangement (see Figure 7 and Appendix A). Regions of both stable flows and highly fluctuating flows co-exist. The highly fluctuating regions are most often connected into streaks of 2–4 stagnant zones at irregular locations within the disordered array. The stagnant zones are characterized by low DNA concentrations and a small degree of cross-flow. A higher degree of fluctuations and periodic blob growth and shedding (as seen in the quadratic array) occurs in the upstream locations within these streaks. However, we also observe stable flow regions with a lack of switching of the flow direction, which is not seen in the quadratic and hexagonal arrays. In between the highly unstable and stable flow regions, there is a spectrum with varying degrees of spatial flow fluctuations depending on the location within the array. The regions seem to last indefinitely and reappear in the same locations after repeatedly turning off and on the pressure that drives the flow.

### 3.4. Microscopic Patterns and Polarization

With the DNA stained with intercalating dyes, we can utilize two-channel polarization microscopy to probe the local orientation of extended DNA molecules (see the high resolution (100×) micrographs in Figure 8). The images are colorized with the pixel value denoting fluorescence intensity and the hue denoting the emission polarization ratio, $P=\left(I_{\|}-I_{⊥}\right)/(I_{\|}+I_{⊥}$). $I_{\|}$ and $I_{⊥}$ refer to the two perpendicular polarization channels placed at a 45-degree angle to the long axis of the array.

We showed in our previous work [11] that the flow shifts orientation locally at the pillars in the quadratic array at high flow velocity. A similar phenomenon is observed with the hexagonal array (see Figure 8). The local orientation and stretching of DNA switches between right and left for both quadratic and hexagonal flows.

At low flow velocities, *P* is close to 0 for the three array geometries, implying little or no polymer directionality. The flow appears laminar with few observed flow instabilities. At higher flow velocities, high levels of elastic instability together with larger values of P are observed. High *P* can be expected as DNA molecules are known to extend during high shear and extensional rates [28,29]. The flow pattern of the quadratic and the hexagonal arrays appear remarkably similar, with switching of the flow direction between left and right (see Figure 8F–H,J–L). Please note that between the switching events, the flow patterns in the two array types differ. In the quadratic array, dark regions of low DNA concentration form in the gaps between the pillars in the flow direction. Periodic cycles of accumulation and shedding of DNA mass are observed in each of the two vortices formed in the pillar gaps (see Figure 8G and Appendix A). For the hexagonal array, the DNA molecules flow along the highly curved paths, exhibiting strong preferential orientation in either of the polarization channels.

### 3.5. The Flow around Sparsely Distributed Pillars Couples Only in the Flow Direction

To illustrate how the proximity of the pillars affects the formation of the waves, we investigate a sparse array, where the pillars are separated from each other by a distance (100 μm) that is large in magnitude compared to the size of the pillars. Interestingly, we do not observe any lateral coupling between the flow of neighboring pillar columns. We note that the distance between the pillars is much greater than the contour lengths of the longest DNA considered in this work (see Table 1), as well as compared to the size of the vortices. Analyzing the space in between the pillars, we find a highly uniform flow of DNA with little fluctuations (see Figure 9A–C). At higher flow velocities, vortex pairs form at the pillars along with the growth of a DNA blob and subsequent shedding of this blob (see Figure 8B–D and Appendix A). The apparent blob size is much larger than in the quadratic or disordered arrays at similar flow velocities. The shedding of a blob destabilizes the downstream vortices of the same array column and leads to a cascade shedding of the DNA blobs downstream (see Figure 9D or more clearly in Appendix A).

## 4. Discussion

### 4.1. Overall Pattern Formation Is Connected to Flow Behavior around Individual Pillars

As is clear from Figure 1, we find a striking difference in the large-scale flow patterns of the quadratic and the hexagonal arrays. Based on the results of polarization imaging at high magnification (see Figure 8) we propose that this is primarily due to the smaller local curvature in the hexagonal array compared to the quadratic one. In the quadratic array, it is the formation of stagnation zones between the pillars in the flow direction, where the maximum curvature is located, that precedes all other effects. In the hexagonal array, while there are stagnation zones, they are considerably smaller. The time required to build up instabilities (vortices) and for them to collapse is, therefore, considerably smaller in the hexagonal array, leading to higher temporal frequencies in the fluctuations (see Figure 5). The amplitude and spatial extent of the flow instabilities in the arrays are seen to increase with flow velocity or De (see Figure 5).

Instabilities occurring upstream of pillars, similar to those that we observe, have been studied by others [16,30,31,32,33]. Although they have also observed growth of the instability as a function of Wi and the wobbling between two configurations, only Kawale et al. have observed the subsequent shedding of such instabilities [31] when studying the flow of hydrolyzed polyacrylamide (HPAM) across multiple geometries. Interestingly, they observed “dead zone washing” in both their quadratic and hexagonal arrays. They found that the hexagonal arrays exhibit smaller dead zones with a “prism”-like shape. It is possible that growth and shedding of DNA also occur in our hexagonal array, but that occurs at spatial and temporal scales that are beyond our experimental capabilities and cannot be ruled out. However, it could also be that, because the dimensions of the arrays and the composition of the viscoelastic fluid used in our work differ from that of Kawale et al., this alters the elastic dynamics significantly and prevents shedding and growth cycles from occurring in our hexagonal arrays. Although Kawale reported that a shedding event disturbs downstream vortices, they did not observe any large-scale flow patterns like the waves that emerge in our system. We hypothesize that the important factors that distinguish our work from that of others are that our arrays are denser and of much smaller array dimensions. The small dimensions lead to much higher *El* and stronger elastic effects, whereas the high pillar density leads to smaller distances between flow patterns and thus allows for a higher level of coupling between dead zones, vortices, and other effects at the individual pillar scale. Other experimental studies on flow across disordered arrays have had much higher porosity (0.84 [17,18]) with larger pillar radii and gaps (*R* = 50 μm, G≈ 95–240 μm [17,18]) and different viscoelastic fluids, e.g., polyacrylamide-based Boger fluids [18] and aqueous wormlike micellar (WLM) solutions [17].

### 4.2. Disordered Arrays Prevent Long-Range Synchronization

The lack of long-range flow coupling in the disordered array points to a fundamental mechanism in the formation of waves. In high magnification images (100×, Figure 7) and in Appendix A, we see that both concentration variations around pillars and vortices in stagnant zones build up in the disordered array. However, because of the random nature of the array, each stagnant zone is different, and the formation and collapse of vortices are also different. There is, therefore, no global population of similar states that can synchronize to form waves. Similar to what others have observed [14,17], we observe multiple flow states across our arrays. Various steady-state patterns form in geometries that we might identify and use as the basis for new designs with specific functions, such as maximizing or minimizing the depletion of DNA locally. Increased disorder in 2D-micropillar-array geometries has been shown to both suppress [18] and enhance flow instability [17]. Both referenced works report varying degrees of flow velocity fluctuations depending on the local pillar distribution and find stagnation zones when pillars are next to each other in the flow direction. In the earlier example, Walkama et al. identified that a disordered array geometry leads to preferential flow paths, which promote shear over extensional flow by reducing the polymer stretching [18]. These preferential paths are similar to those that we observe in the disordered array. Conversely, in the latter example, Haward et al. explained their finding of enhanced flow fluctuations due to the occurrence of stagnation points, which leads to high tensile stresses and elastic instabilities [17]. In our system, we observe a highly reduced level of flow instabilities for the case of a disordered geometry, consistent with the findings of Walkama et al. [18].

We want to stress that our work does not represent a comprehensive study of disordered arrays. There are multiple ways of designing a disordered array, which may lead to significantly different flow dynamics. The pillars in our disordered design are still located along periodic rows, even if the rows are laterally shifted with respect to each other. A completely randomized pillar distribution or larger variation in the gaps with overlapping pillars (i.e., pillars of various sizes) could result in an entirely different flow pattern. See, e.g., the disordered array design of De et al. [34]. We believe that factors such as the pillar radius, porosity, and the properties of the viscoelastic fluid (relaxation time, contour length to pillar radius and array pitch ratios, and the degree of shear thinning) all play an important role in forming the flow pattern. It, therefore, becomes complicated to compare our work directly to that of others.

### 4.3. Flow Patterns around the Sparsely Positioned Pillars Are Synchronized in the Flow Direction but Not Laterally

We show similar behavior in our DNA solutions that others have reported for other types of viscoelastic fluids around pillars of varying dimensions. We find instabilities both upstream and downstream of the pillars in our sparse arrays. As seen by some authors [31,35], we also observe that the upstream instabilities collapse or shed at high flow velocities (see Figure 8 and Figure 9). When the vortices upstream of the pillars shed DNA blobs, these blobs persist for long enough that they trigger the collapse of vortices far downstream. However, no lateral wave behavior is observed for the array of sparsely distributed pillars. A simple explanation is that even if vortices form and shed in the vicinity of the pillars, they are too far apart to communicate laterally. Our results are in alignment with Shi et al. [13], who found that a smaller spacing between pillars in a 1D array leads to stronger flow fluctuations. Although they do not report any vortex shedding, they do observe that instability immediately downstream of one pillar affects the upstream instability of a pillar further downstream. We notice that the blobs of the sparse array are much larger than those in the quadratic or disordered arrays. We believe that this is due to the lack of confinement that occurs in the dense arrays by the tightly placed pillars. The lack of interacting neighboring vortices could also allow for the formation of larger blobs before collapse is triggered.

We observe that there is no lateral interaction in the sparse array, and we propose that the coupling here depends on distance only. Conversely, in the flow direction (row-to-row), we propose that the interaction depends on both distance and time. For the distance dependence, we compared the contour length of the DNA with the row-to-row distance and found that for DNA shorter than the row-to-row distance, we would not see any wave formation [11]. For the time dependence, we need to consider the time it takes for a given fluid element to go between two pillars, which in turn is related to the row-to-row distance and the flow speed, and relate that to the lifetime of the relevant flow perturbation. The latter can be expressed as a Deborah number with relaxation time corresponding to anyone out of several time scales, such as the polymer relaxation time or the vortex lifetime.

### 4.4. Cyclic and Random Large-Scale Fluctuations

The fluctuations exhibit a wide range of random and cyclic behaviors. We highlight various perspectives that we believe will be valuable for future theoretical investigations of the phenomenon.

The high *coherence* of the waves in the quadratic array as opposed to the hexagonal array (see Figure 1), is made clear by the kymographs in Figure 2. Here, we can see that the waves are much longer-lived for the quadratic array than for the hexagonal array.

*Flow rate* has a strong influence on the fluctuations. We observe that the fluctuations increase with increasing flow rate. In line with what has been reported in the literature, we also see that the frequency of the fluctuations increases with increasing flow rates (see Figure 5). Qin et al. observed that an elastic wave propagating upstream from a cylinder increases with velocity and penetrates higher upstream with higher *Wi* [32]. Varshney and Steinberg observed an increase in the peak frequency and the corresponding cross-stream elastic “wave” velocity with increased *Wi* [33]. Others have seen that temporal velocity fluctuations often increase in magnitude with increased *Wi*. This has been the case for many geometries, including a 1D pillar array [36], ordered 2D pillar arrays [17,18,37], and disordered 2D pillar arrays [17,18]. Spatial velocity fluctuations have also been shown to increase in magnitude with increased *Wi* for ordered 2D pillar arrays [18] and disordered 2D pillar arrays [18].

The *standard deviation* can be used to distinguish the different wave patterns that we observe. Remarkably, the relationship between the maximum standard deviation as a function of average pixel value seems to be linear with different slopes for the three different array geometries, and it varies with flow rate, indicating that it originates in the characteristic fluctuations in each geometry. Conversely, while the maximum average is also linearly related to the standard deviation, it has similar slopes for quadratic, hexagonal, and disordered arrays, and it is independent of flow rate, implying that it originates in imaging noise.

We report clear trends in the *scaling relationships* of the amplitude versus frequency plots for the different geometries. At low flow rates where no waves are observed, the exponent is similar for all array types. In contrast, for high flow rates and for the quadratic and hexagonal arrays where the waves appear, the exponent gradually changes to more negative values as the flow rate is increased. We note that in the absence of waves, the noise is consistent with a 1/f power spectrum. Compared to the literature for viscoelastic fluctuations, we need to keep in mind that most power laws are derived from velocity fluctuations and not concentration fluctuations, as in our case. We display our results as the amplitude, which means that we need to multiply our exponents by two to be comparable with the power density exponents. Therefore, we have approximately −1 for the power law exponent for the disordered array and the case of no waves. Consistent with our findings, we notice that the power law exponents in most cases are more negative than the Kolmogorov exponent of −5/3 that is associated with high-Re inertial turbulence. We can also note that just like in our case, the power law exponents depend on the exact geometry considered, with approximately −3 for the quadratic array and approximately −4 for the hexagonal array. Groisman and Steinberg found the power law decay exponent to be −3.5 [1]. Pan et al. [36] measured the power law decay exponent to be −1.7 immediately following a 1D array and −2.7 far downstream. De et al. measured the power law decay exponent of streamwise and lateral velocity fluctuations in a 2D pillar array to be approximately −3 [37]. Grilli et al. used Lagrangian simulations based on an Oldroyd-B constitutive equation for a 1D pillar array and found the power law decay exponents to be −3.4 and −4.3 for various locations around a cylinder. Haward et al. found the power law decay exponent to be −3.7 (staggered) and −2 (aligned) at low *Wi* (*Wi* 7.5) and −2.2 at high *Wi* 75 [17]. Qin and Arratia measured a power law decay exponent of −1.7 just after a 1D pillar array and −2.7 far downstream [38]. Ekanem et al. found that the power law decay exponent was up to −2.1 at high pressure fluctuations [39]. Browne et al. found the power law decay exponents in their 3D porous media to vary from −1.1 to −1.4 and −0.8 to −1.1 at *Wi* 3.9 and 4.4, respectively [10]. Browne refers to 11 other papers with varying exponent magnitudes, all between −1 and −4.6 or −1 and −3. According to the theoretical work by Fouxon and Lebedev, the power law decay exponent should be more negative than −3 [40], which is consistent with some experimental results, however, with several exceptions.

## 5. Conclusions

We have shown that changing the spatial pillar distribution transforms the large-scale fluctuations in a λ DNA solution at high concentration as it flows across a micropillar array. In contrast to the ordered regular waves seen for the quadratic array [11], we see seemingly chaotic large-scale fluctuations for the hexagonal array, and we note an absence of fluctuations when slightly randomizing the array pattern.

In a recent publication [23], we show that the characteristics of the waves can be determined to a large part by the shape of the pillars. In the paper, we also showed that the waves can be suppressed by alternating the pillar geometry such that the necessary conditions for the waves cannot be reached. From a different perspective, we here posit that a necessary condition for the formation of waves is that the microscopic flow patterns at each pillar are synchronized across the device. One way to prevent synchronization is to break translational symmetry in the device by changing the order in which the pillars are arranged. Indeed, we see a clear suppression of the waves for the randomized array. Similarly, we note that if the distance between pillars is large, i.e., in the sparse array, we do not see any correlation between the flow behaviors at the individual pillars and thus no formation of waves. The outcome is consistent with what we see for the effect of the relation of the contour length of the DNA to the row-to-row distance in our previous work [11]. We conclude that to fully understand the role of the distances between pillars in the formation of the waves, we need to take into account the time it takes for a flow perturbation to go from one row to the next, the relationship between the polymer contour length and relevant pillar-to-pillar distances as well as the time scales related to polymer relaxation and vortex dynamics. The importance of the long-range synchronization is also illustrated by the fact that while on the microscale, the quadratic and hexagonal arrays feature striking similarities, with rapid switching in the directions downstream of every pillar, the different translational symmetry of the array results in dramatically different wave patterns as well as preferred flow paths.

This work and our previous work [11,22,23], all explore variations of the design that take place in the plane of the device, i.e., in a 2D context with weak confinement where the ratio of the pillar size and the device depth is O(1). With both inertial turbulence [41] and viscoelastic fluctuations [42,43,44,45] exhibiting flow patterns that depend on dimensionality, a natural next step would be to investigate the effect of vertical confinement. A very thin device would constrain the movement of any vortices, which in turn would be expected to have a significant effect on the formation of the waves. Strong confinement in one dimension is also known to increase the relaxation time of the DNA [46] and to decrease its spring constant [47], which is consistent with the behavior of the molecule in one-dimensional nanochannels of moderate confinement at the scale of the persistence length and larger (the deGennes regime) but opposite to the behavior in strong confinement at a scale less than the persistence length (the Odijk regime) [48]. We should, therefore, expect a non-monotonous dependency of the Deborah number on the device depth for a given flow rate, which in turn could be used to control which flow rates lead to wave formation. We conclude that it would be highly interesting to explore the degree of confinement through various depths of the device in relationship to DNA persistence length, DNA blob size, i.e., the radius of gyration, the size of the vortices, and typical lateral length scales in the device.

Although our work concerns the dynamics of the fluid in a fixed array, another possibility that has been explored by several groups is to also include the dynamics of flexible pillars that give additional degrees of freedom for engineering applications. Here, a recent example of canopy waves resembles our waves, except they are observed as a movement of the pillars rather than fluctuations in concentration [49]. It would be highly interesting to explore these types of waves for different geometries, degrees of order, and symmetry, just like we demonstrate here and have demonstrated previously [11,23].

Our work connects to several other fields of complex systems, opening up new investigations. We see a clear order/disorder transition. This applies to both the overall behavior of the flow patterns as the local ordering of the DNA, forming a nematic liquid crystal phase as revealed by the polarization data and showing similarities, for example, with early work by Livolant and coworkers in Refs. [50,51], with broader relevance to molecular crowding of DNA solutions [52,53]. The time-averaged data reveal characteristic persistent patterns. In light of the concentration waves moving through the array, these could potentially be the result of interfering counter-propagating waves, a further deeper study of which might elucidate deeper mechanisms of the waves.

Our experimental findings can aid in the fundamental understanding of elastic flow through porous media. By altering the pore structure geometry, the flow dynamics changes, which could be exploited in important applications by e.g., enhancing or suppressing flow mixing in microfluidic devices with relevance for transport of complex fluids, sorting of biomedical samples and enhanced chemical reactions by controlled mixing of reagents. Specifically, for mixing applications, the waves have been shown to lead to mixing [23] consistent with what has been shown for elastic turbulence [5,54]. For sorting and unmixing applications, one approach is to suppress the waves by carefully leveraging the effects of symmetry as we reported recently [23], or by randomizing the array sufficiently to suppress the waves, but not excessively such that the sorting mechanism would be hindered. Finally, it may be conceivable to utilize the waves for turbophoretic unmixing [55,56]. However, this would require a better understanding of not only the flow of the DNA but also of the flow of the solvent.

## Figures and Tables

**Figure 1 micromachines-15-00268-f001:**
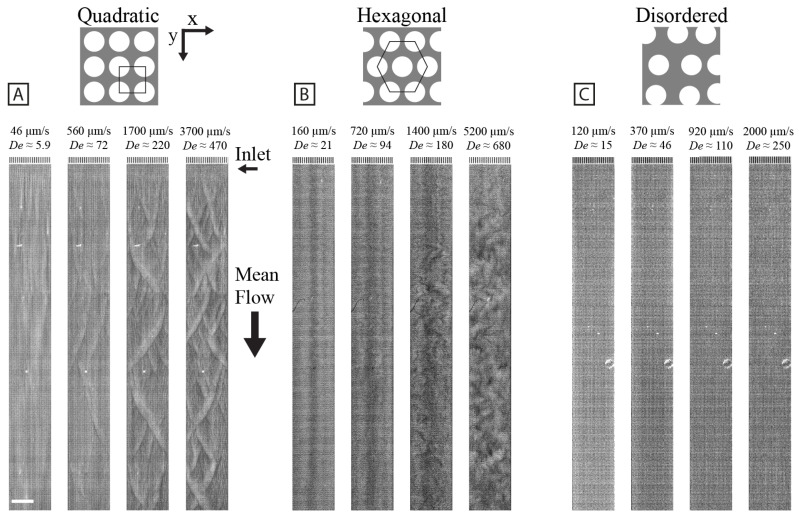
Large-scale flow pattern as a function of array geometry and flow velocity. Repetitive wavefronts can be seen in the quadratic array at higher flow velocities (**A**), whereas smaller repetitive zig-zag patterns are visible in the hexagonal array (**B**). In contrast, no large-scale flow pattern is visible in the disordered array (**C**). Please note that the vertical stripes seen in the lower flow velocity micrographs and in the disordered array are most likely a combination of sample concentration fluctuations together with viscoelastic focusing occurring in the inlet channels. The brightness and contrast settings are the same for all micrographs of the same array type. Please note that some fabrication defects are visible, i.e., the bright specks in (**A**), the black strand in (**B**), and the bright ring in (**C**). The devices are made with a pillar radius of R≈7μm. The quadratic and hexagonal arrays are made with a pillar–pillar gap of G≈4μm, while for the disordered array, the average pillar–pillar gap is G≈4.5μm. Detailed information about the device dimensions for the different designs is given in Table 1. The magnification is 2×, and the exposure time for all micrographs is 30 ms. Scale bar is 400 μm. The data in (**A**,**C**) have been adapted from [11], DOI: 10.1039/D2LC01051H, under the terms of the CC BY 3.0 license https://creativecommons.org/licenses/by/3.0. See Appendix A for a video representing the raw data.

**Figure 2 micromachines-15-00268-f002:**
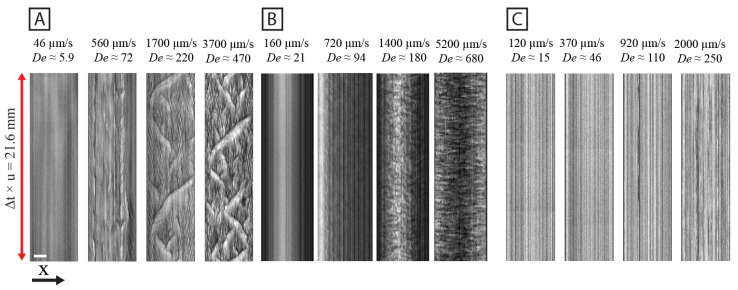
Kymographs of quadratic (**A**), hexagonal (**B**) and disordered (**C**) arrays for increasing flow velocity. The kymographs are based on lines perpendicular to the long axis of the array in between two rows of pillars. The vertical lengths are normalized so that the flow velocity multiplied by the time span is equal for all kymographs. The brightness and contrast are set so that they are the same across the flow velocities for each array type. The horizontal scale bar is 200 μm. The data are based on low magnification (2×) videos, same as is presented in Figure 1.

**Figure 3 micromachines-15-00268-f003:**
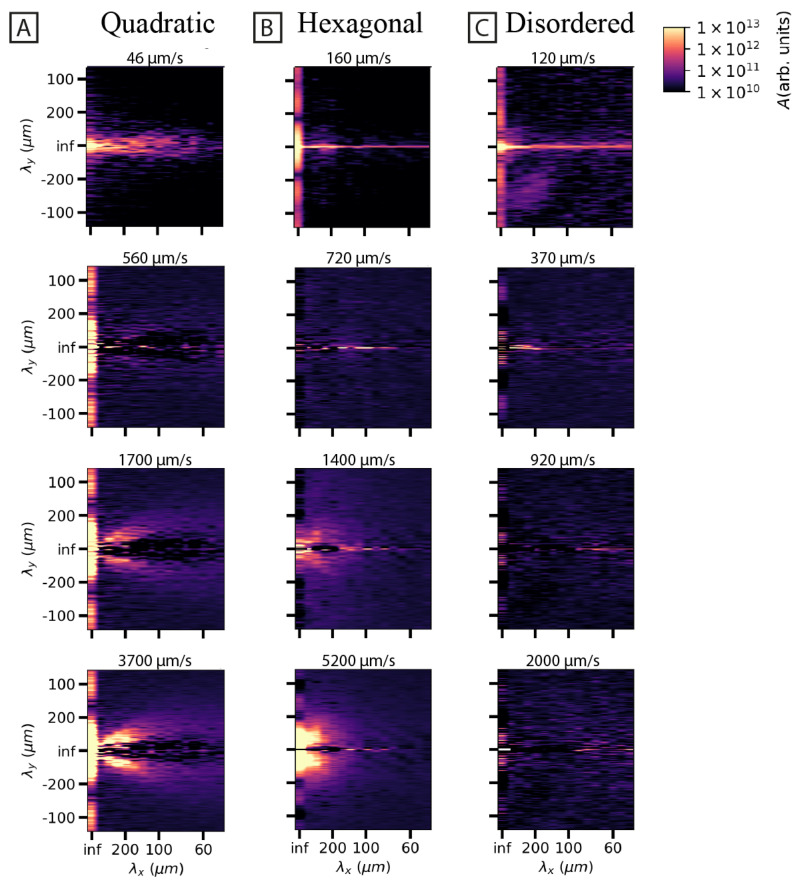
Logarithmically scaled and time-averaged two-dimensional Fourier spectra of DNA sample flows across quadratic (**A**), hexagonal (**B**), and disordered (**C**) array patterns as a function of mean flow velocity across the device. The bottom three rows of spectra for the array patterns have had the top row (background) subtracted to highlight the discrepancy between the arrays. The peak frequencies within the quadratic array at high pressure are limited to a narrow cone with wavelengths within approximately 300 μm parallel with the flow direction (y) and approximately 150 μm perpendicular to the flow (x). Instead, the hexagonal array shows a high signal at similar low frequencies for the entire range of angles. The panels show only the positive wavelengths in the x-direction (perpendicular to the flow direction) and not the negative ones. The high-magnitude peak at infinite wavelength corresponds to the zero-frequency component, while the high-magnitude horizontal line corresponds to the pillar rows. The image brightness and contrast have been set so that the absolute amplitude limits are the same for all the array patterns and pressure runs. The spectra are based on data recorded during 30 s at low magnification (2× objective), same as is presented in Figure 1.

**Figure 4 micromachines-15-00268-f004:**
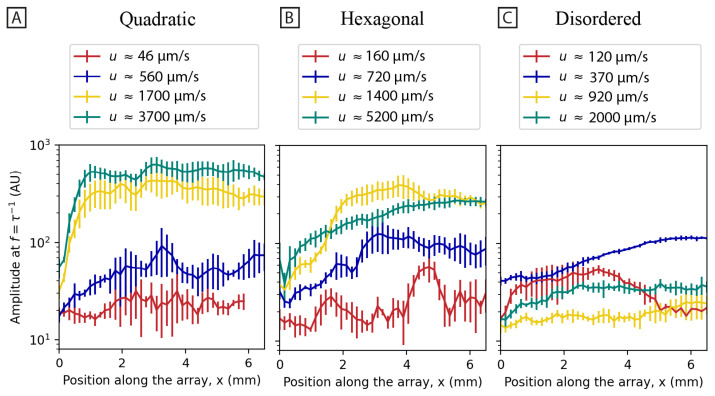
Strength of the waves as a function of position along the array in the devices. Temporal Fourier amplitudes are plotted as a function of position along the quadratic (**A**), hexagonal (**B**), and disordered (**C**) arrays. Temporal Fourier transforms along each row of regions of radius 81.5 μm are averaged (in total, 41 rows with 5 regions in each row (see Appendix A)). The amplitudes are averaged over a factor of two in frequency, i.e., at the frequencies $\sqrt{(}2)\tau^{-1}$ to $\sqrt{(}{2)}^{-1}\tau^{-1}$ where τ is the relaxation time of the DNA for each position along the channel. The error bars denote the standard deviation of the Fourier amplitudes among the regions in each row. Graphs are plotted for different flow velocities. The data are based on 30 s videos recorded with 2× magnification (same as is presented in Figure 1).

**Figure 5 micromachines-15-00268-f005:**
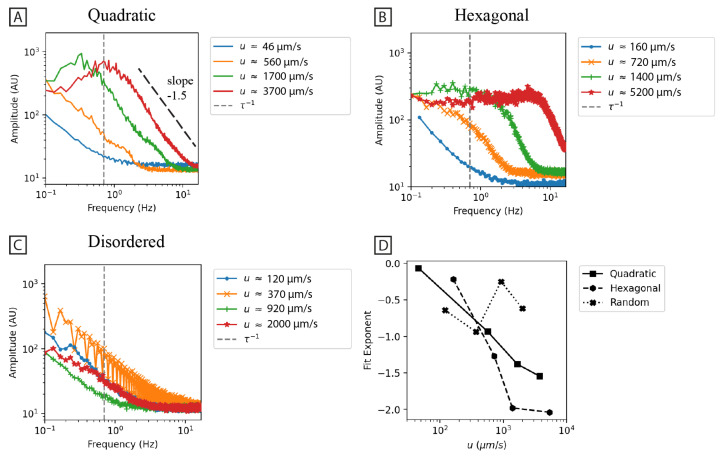
Fourier transforms of temporal variations. (**A**–**C**) show the average 1D Fourier amplitude spectra for 30 s recordings of the quadratic, hexagonal, and disordered arrays at 2× magnification (same data as is presented in Figure 1). The averages are based on the mean intensities of 5 × 41 circular regions of radius 81.5 μm across the entire field of view. (**D**) The amplitude spectra are analyzed by first identifying the knee in the spectra and subsequently fitting an exponential to the sloped region immediately to the right of the knee. The exponent is plotted as a function of applied mean flow velocity. The inverse relaxation time, $\tau^{-1}$, has been outlined in (**A**–**C**).

**Figure 6 micromachines-15-00268-f006:**
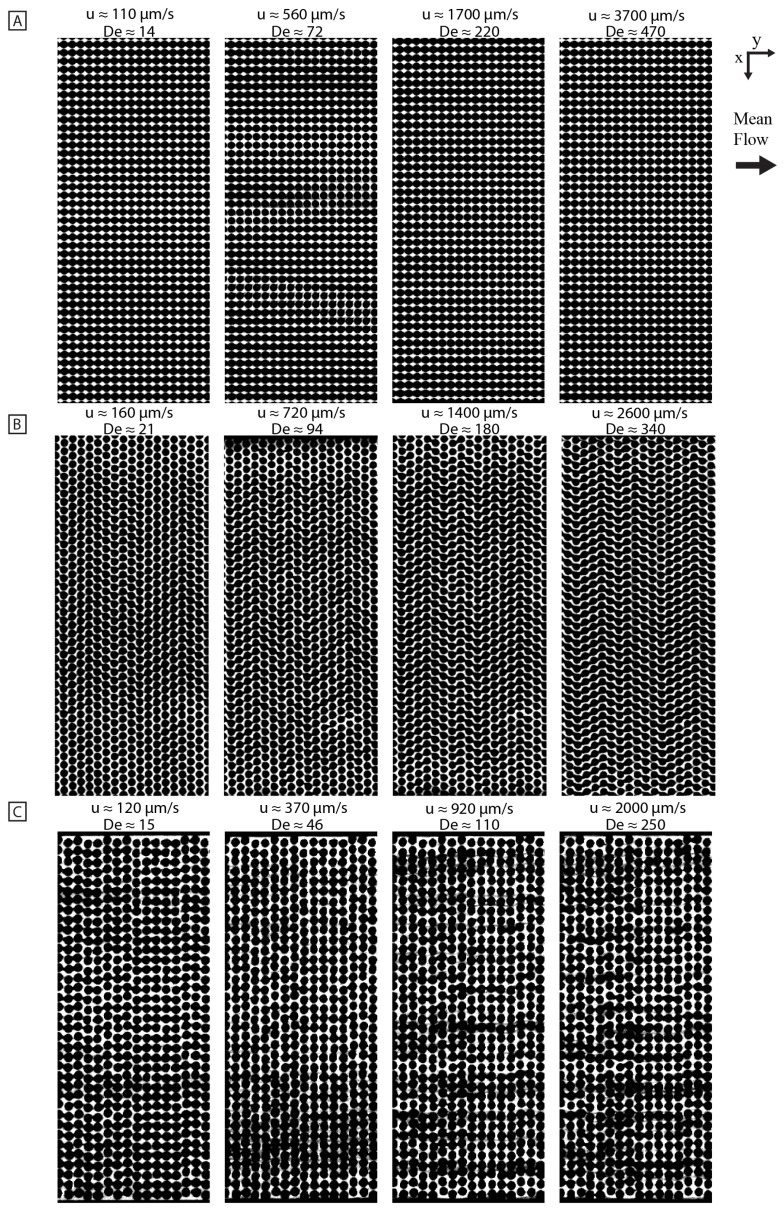
Time averages (30 s) for the three dense pillar arrays: quadratic (**A**), hexagonal (**B**) and disordered (**C**). The brightness and contrast have been set so that the features are enhanced for each image. The exposure time for all recordings was 30 ms. The frame rates were 2 fps, 12.5 fps, 21 fps, and 33 fps for the images left to right in (**A**) and 2 fps, 12.5 fps, 21 fps, and 21 fps for the images left to right in (**B**,**C**). The scale bar is 100 μm, and the data are based on 4× magnification measurements.

**Figure 7 micromachines-15-00268-f007:**
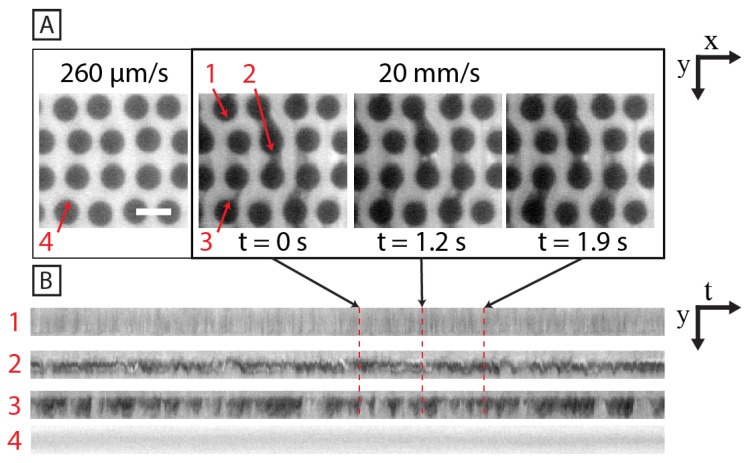
The disordered array exhibits high local flow pattern variation. (**A**) Micrograph snapshots of the disordered array at low and high pressures, emphasizing the wide variation in local flow behavior. The red arrows show (1) a highly stable flow region at high DNA concentration. (2) A region with a vortex pair, continuously generating and shedding blobs. (3) An unstable flow region that switches from low to high DNA concentration and low to high flow velocity. (4) An example region showing high stability at low flow velocity. (**B**) Kymographs of the row-gap areas marked out with red arrows in (**A**). Scale bar is 20 μm in (**A**). The data have been recorded with 20× magnification. See also Appendix A for a dynamic depiction of the high-velocity flow.

**Figure 8 micromachines-15-00268-f008:**
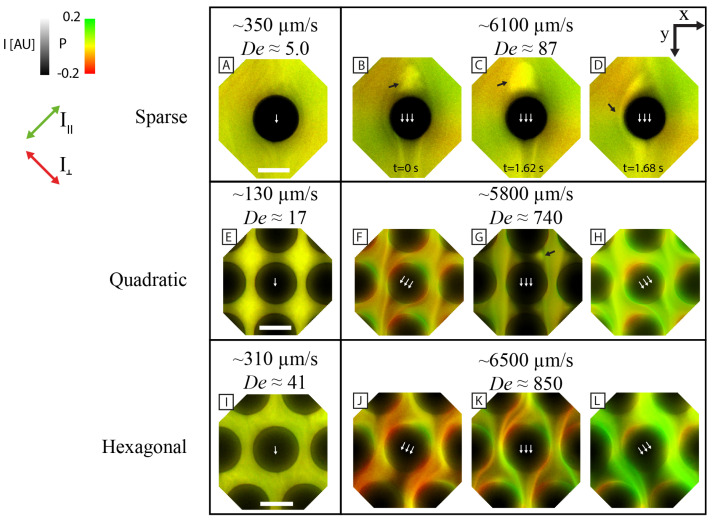
Polarization micrographs (100 × objective) of polymer orientation on the scale of individual pillars. There is a stark difference in the small-scale dynamics between the sparse (**A**–**D**), quadratic (**E**–**H**) and hexagonal (**I**–**L**) arrays. The images are composed of the hue, saturation, and value (HSV) color model, where the fluorescence intensity is denoted by the pixel value and polarization ratio by the color. Please note that the colored arrows to the left indicate the two polarization directions that, in turn, are perpendicular to each corresponding orientation of the DNA. At low flow rates (**A**,**E****,I**), the flow direction is parallel to the long axis of the array for all array types. In the sparse and quadratic (and disordered arrays), the following phenomenon is observed: A mass of DNA strands (a blob) grows and subsequently is shed in the vortex pair upstream of the pillar. The blobs are denoted with black arrows. At high flow rate in both the quadratic (**F**–**H**) and the hexagonal array (**J**–**L**), the DNA strand flow fluctuates in the direction between the left or the right of the long axis of the array. Please note that the DNA strands are concentrated into diagonal streaks and that there is a significant number of extended polymers that are stuck at the trailing edge of the pillars, resulting in a contrasting polarization ratio. The scale bars correspond to 10 μm. Please note that the pillars are slightly laterally compressed at higher flow velocities. Subpanels (**E**–**H**) have been reproduced from [11], DOI: 10.1039/D2LC01051H, under the terms of the CC BY 3.0 license https://creativecommons.org/licenses/by/3.0, and the Deborah numbers have been recalculated according to the definition used in this article.

**Figure 9 micromachines-15-00268-f009:**
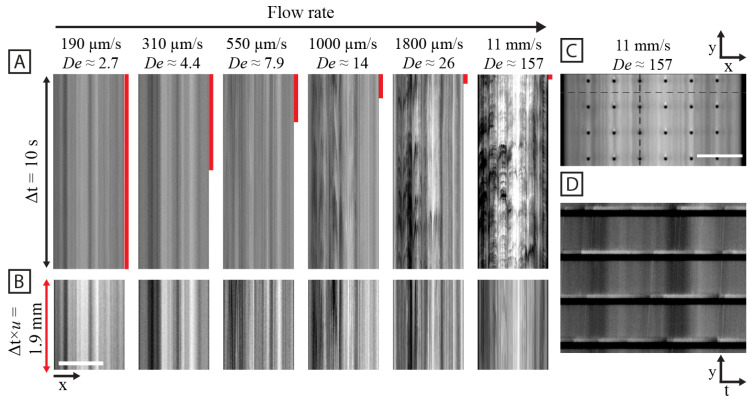
The flow across the array of sparsely distributed pillars is overall homogenous. However, vortices form upstream of the pillars where DNA is periodically collected and subsequently shed. A shedding event triggers the downstream vortices to shed. (**A**) Kymographs of the dashed horizontal line in (**C**) showing the dynamics that appear at high De. (**B**) Kymographs with the same data from (**A**) where the time span is selected so that the traversed distance is kept constant. The relative durations for the kymographs in (**B**) are plotted as red bars in (**A**) for respective flow velocity. (**C**) Fluorescent micrograph snapshot at high flow velocity. Note the upstream DNA blobs. (**D**) Kymograph of the dashed vertical line in (**C**) to elucidate the cycles of growth and DNA blob shedding. The blobs (bright) grow in time until they are shed (dark section). The devices are made with a pillar radius of R≈7μm. The pillar–pillar gap is G≈100μm. Detailed information about the device dimensions is given in Table 1. The kymograph spans a total of 6.4 s. Scale bars are 500 μm in (**B**) and 200 μm in (**C**).

**Table 1 micromachines-15-00268-t001:** Overview of the device dimensions *. Depth of all devices is h=10.9±0.1μm.

Array Pattern	Pillar Radius r **(** μ **m)**	$N_{x}$ ^1^, $N_{y}$ ^2^	Porosity ^3^	Lateral Pillar Gap G **(** μ **m)**	Lateral Gap-to-Contour-Length Ratio ^4^ **(** G/L **)**
Quadratic	7.1±0.4	44, 444	0.52	4.1 ± 0.4	0.25
Hexagonal	7.0±0.2	44, 444	0.52	3.9 ± 0.2	0.25
Disordered	7.0±0.1	37, 381	0.54 ± 7.5	4.5 ± 1.5	0.27 ± 0.09
Sparse	7.0±0.1	6, 70	0.99	100	6.06

^1^ Number of pillars in the lateral direction. ^2^ Number of pillars in the longitudinal direction. ^3^ Porosity is defined as the volume ratio of the pores to the entire volume of the channel. ^4^ Lateral gap-to-contour-length ratio is based on a contour length of λ-phage DNA of 16.5 μm. * The ranges that are given for the entries represent the measurement uncertainty except for the porosity and lateral gap-to-contour ratio, where they represent the variation in the actual values across the pillar array in the disordered device.

## Data Availability

The data that support the findings of this study are openly available in Harvard Dataverse at https://doi.org/10.7910/DVN/IRRX1P (accessed on 19 October 2023).

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
