# Peer review of "Geometry-Dependent Elastic Flow Dynamics in Micropillar Arrays"

_micromachines, 2024, doi:10.3390/mi15020268_

Round 1

Reviewer 1 Report

Comments and Suggestions for Authors

investigates the flow dynamics in micropillar arrays with different array displacements, focusing on how these distribution differences affect large-scale flow patterns and DNA wave formations. The study examines quadratic, hexagonal, and disordered array distribution, using high DNA concentration solutions. It reveals that array distribution significantly influences the flow dynamics. Overall, this work shows an interesting angle of the complex flow and its potential application in DNA sorting. I support the publication of this work after addressing the a few comments below:

1. Can the authors give a discussion of how the distance between different pillar rows affects the wave pattern? Also, if the height of the pillars affect the wave pattern?

Author Response

  1. Can the authors give a discussion of how the distance between different pillar rows affects the wave pattern? Also, if the height of the pillars affect the wave pattern?

RESPONSE:   These are indeed valuable and important points that connect to the bigger question about the connection of length scales in the design to the behavior of the waves. Note that we have already presented evidence in a previous publication that the length of the DNA must be greater than the row-to-row distance for the waves to form[1].

**** Concerning the effect of the row-to-row distance, we have updated section 2, Methods, accordingly along with the figures 8 and 9

The Deborah number is now consistently defined throughout the manuscript based on the array pitch. The text in the last paragraph of the Methods section has been updated accordingly. In addition, Deborah numbers given in figures 8 and 9 have been updated accordingly.

**** Concerning the effect of the row-to-row distance, we have added a few sentences to section 4.3, Discussions, as follows:

We observe that there is no lateral interaction in the sparse array and we propose that the coupling here depends on distance only. Conversely, in the flow direction (row-to-row) we propose that the interaction depends on both distance and time.  For the distance dependence we compared the contour length of the DNA with the row-to-row distance and found that for DNA shorter than the row-to-row distance, we would not see any wave formation [1]. For the time dependence we need to consider the time it takes for a given fluid element to go between two pillars, which in turn is related to the row-to-row distance and the flow speed, and relate that to the lifetime of the relevant flow perturbation. The latter can be expressed as a Deborah number with relaxation time corresponding to anyone out of several time scales, such as the polymer relaxation time or the vortex lifetime.

**** Concerning the effect of the row-to-row distance, we have added a few sentences to section 5, the Conclusions, as follows:

The outcome is consistent with what we see for the effect of the relation of the contour length of the DNA to the row-to-row distance in our previous work [1]. We conclude that to fully understand the role of the distances between pillars in the formation of the waves, we need to take into account the time it takes for a flow perturbation to go from one row to the next, the relationship between the polymer contour length and relevant pillar-to-pillar distances as well as the time scales related to polymer relaxation and vortex dynamics.

**** Concerning the effect of the height of the pillars, we have added a few sentences to section 5, the Conclusions, as follows:

This work and our previous work [1-3], all explore variations of the design that take place in the plane of the device, i.e. in a 2D context with week confinement where the ratio of the pillar size and the device depth is O(1). With both inertial turbulence [4] and viscoelastic fluctuations [5-8] exhibiting flow patterns that depend on dimensionality, a natural next step would be to investigate the effect of vertical confinement. A very thin device would constrain the movement of any vortices, which in turn would be expected to have a significant effect on the formation of the waves. Strong confinement in one dimension is also known to increase the relaxation time of the DNA [9] and to decrease its spring constant[10], which is consistent with the behavior of the molecule in one-dimensional nanochannels of moderate confinement at the scale of the persistence length and larger (the deGennes regime) but opposite to the behavior in strong confinement at a scale less than the persistence length (the Odijk regime)[11]. We should therefore expect a non-monotonous dependency of the Deborah number on the device depth for a given flow rate, which in turn could be used to control which flow rates lead to wave formation. We conclude that it would be highly interesting to explore the degree of confinement through various depths of the device in relationship to DNA persistence length, DNA blob size, i.e. the radius of gyration, the size of the vortices, and typical lateral length scales in the device.

[1]       Ström, O.E., J.P. Beech, and J.O. Tegenfeldt, Short and long-range cyclic patterns in flows of DNA solutions in microfluidic obstacle arrays. Lab on a Chip, 2023. 23(7): p. 1779-1793. 10.1039/D2LC01051H

[2]       Beech, J.P., O.E. Ström, E. Turato, and J.O. Tegenfeldt, Using symmetry to control viscoelastic waves in pillar arrays. RSC Advances, 2023. 13(45): p. 31497-31506. 10.1039/D3RA06565K

[3]       Ström, O.E., J.P. Beech, and J.O. Tegenfeldt High-Throughput Separation of Long DNA in Deterministic Lateral Displacement Arrays. Micromachines, 2022. 13,  DOI: 10.3390/mi13101754.

[4]       Boffetta, G. and R.E. Ecke, Two-Dimensional Turbulence. Annual Review of Fluid Mechanics, 2012. 44(1): p. 427-451. 10.1146/annurev-fluid-120710-101240

[5]       McKinley, G.H., R.C. Armstrong, and R.A. Brown, THE WAKE INSTABILITY IN VISCOELASTIC FLOW PAST CONFINED CIRCULAR-CYLINDERS. Philosophical Transactions of the Royal Society of London Series a-Mathematical Physical and Engineering Sciences, 1993. 344(1671): p. 265-304. 10.1098/rsta.1993.0091

[6]       De, S., S.P. Koesen, R.V. Maitri, M. Golombok, J.T. Padding, and J.F.M. van Santvoort, Flow of viscoelastic surfactants through porous media. AIChE Journal, 2018. 64(2): p. 773-781. https://doi.org/10.1002/aic.15960

[7]       Cadot, O. and S. Kumar, Experimental characterization of viscoelastic effects on two- and three-dimensional shear instabilities. Journal of Fluid Mechanics, 2000. 416: p. 151-172. 10.1017/S0022112000008818

[8]       Datta, S.S., A.M. Ardekani, P.E. Arratia, A.N. Beris, I. Bischofberger, G.H. McKinley, J.G. Eggers, J.E. López-Aguilar, S.M. Fielding, A. Frishman, M.D. Graham, J.S. Guasto, S.J. Haward, A.Q. Shen, S. Hormozi, A. Morozov, R.J. Poole, V. Shankar, E.S.G. Shaqfeh, H. Stark, V. Steinberg, G. Subramanian, and H.A. Stone, Perspectives on viscoelastic flow instabilities and elastic turbulence. Physical Review Fluids, 2022. 7(8): p. 080701. 10.1103/PhysRevFluids.7.080701

[9]       Bakajin, O.B., T.A.J. Duke, C.F. Chou, S.S. Chan, R.H. Austin, and E.C. Cox, Electrohydrodynamic stretching of DNA in confined environments. Physical Review Letters, 1998. 80(12): p. 2737-2740.

[10]     Lin, J., F. Persson, J. Fritzsche, J.O. Tegenfeldt, and O.A. Saleh, Bandpass Filtering of DNA Elastic Modes Using Confinement and Tension. Biophysical Journal, 2012. 102: p. 96-100.

[11]     Reisner, W., K.J. Morton, R. Riehn, Y.M. Wang, Z.N. Yu, M. Rosen, J.C. Sturm, S.Y. Chou, E. Frey, and R.H. Austin, Statics and dynamics of single DNA molecules confined in nanochannels. Physical Review Letters, 2005. 94(19): p. 196101.

Reviewer 2 Report

Comments and Suggestions for Authors

The author presents the wave patterns of DNA in pillar arrays under two regular arrangements and a disordered arrangement. This study investigates the differences in wave patterns at both macro and microscopic scales, and establishes a correlation between microscale flow and macroscale waves. This work shows a significant achievement and demonstrates that differences in DNA wave patterns in different array structures can be applied to the transport, separation, and mixing of complex fluids. The manuscript is informative and well-written. However, the following points should be addressed before the acceptance for publication:

1. Could the fabrication defects in Figure 1 potentially lead to variations in DNA density distribution or impact wave patterns?

2. To enhance reader convenience, specific structural information should be provided in figure captions same as Fig.7 in Ref. 11, e.g., Fig.S1, 

3. The velocity units should be consistent, e.g., Fig.1 μm/s and Fig.3 m/s.

4. Line 114. The capitalization format of heading 3.1 appears distinct from the others.

5. Which type of structure array may have an advantage for sorting complex fluids, and how can the target be obtatined?

Comments on the Quality of English Language

I am not qualified to assess the quality of English in this paper.

Author Response

  1. Could the fabrication defects in Figure 1 potentially lead to variations in DNA density distribution or impact wave patterns?

RESPONSE:   While we did not observe any significant effect on the wave pattern of the defects in fig 1, we have observed that certain defects in the arrays, e.g. local deposition of particles, may nucleate wave formation. So far, we have only spurious data on this type of behavior so that we plan instead to publish a careful study of how the waves are influenced by well-controlled perturbations in the arrays.

  1. To enhance reader convenience, specific structural information should be provided in figure captions same as Fig.7 in Ref. 11, e.g., Fig.S1, 

RESPONSE:   We have updated the captions of figures S1 and S2 in the ESI. We have also updated the captions in figures 1 and 9 of the main text. To simply for the reader to obtain a quick idea of the design parameters, we give overall approximate numbers and refer the reader to table 1 in the main text for detailed information.

  1. The velocity units should be consistent, e.g., Fig.1 μm/s and Fig.3 m/s.

RESPONSE:   We have changed the units in the figures such that we use µm/s and no exponents in all figures.

  1. Line 114. The capitalization format of heading 3.1 appears distinct from the others.

RESPONSE:   The capitalization of the headings is now consistent throughout the manuscript.

  1. Which type of structure array may have an advantage for sorting complex fluids, and how can the target be obtatined?

RESPONSE:   **** Concerning choice of structure for sorting applications, we have added a few sentences to the end of section 5, the Conclusions, as follows:

Specifically, for mixing applications, the waves have been shown to lead to mixing [1], consistent with what has been shown for elastic turbulence [2, 3]. For sorting and unmixing applications, one approach is to suppress the waves by carefully leveraging the effects of symmetry as we reported recently [1], or by randomizing the array sufficiently to suppress the waves, but not excessively such that the sorting mechanism would be hindered. Finally, it may be conceivable to utilize the waves for turbophoretic unmixing [4, 5]. However, this would require a better understanding of not only the flow of the DNA but also of the flow of the solvent.

  1. Minor editing of English language required

RESPONSE:   We have carefully gone through the text to improve any deficiencies in the English.

[1]       Beech, J.P., O.E. Ström, E. Turato, and J.O. Tegenfeldt, Using symmetry to control viscoelastic waves in pillar arrays. RSC Advances, 2023. 13(45): p. 31497-31506. 10.1039/D3RA06565K

[2]       Groisman, A. and V. Steinberg, Efficient mixing at low Reynolds numbers using polymer additives. Nature, 2001. 410(6831): p. 905-908.

[3]       Afik, E. and V. Steinberg, On the role of initial velocities in pair dispersion in a microfluidic chaotic flow. Nature Communications, 2017. 8(1): p. 468. 10.1038/s41467-017-00389-8

[4]       Garg, H., E. Calzavarini, G. Mompean, and S. Berti, Particle-laden two-dimensional elastic turbulence. The European Physical Journal E, 2018. 41(10): p. 115. 10.1140/epje/i2018-11726-4

[5]       Esmaily, M., L. Villafane, A.J. Banko, G. Iaccarino, J.K. Eaton, and A. Mani, A benchmark for particle-laden turbulent duct flow: A joint computational and experimental study. International Journal of Multiphase Flow, 2020. 132: p. 103410. https://doi.org/10.1016/j.ijmultiphaseflow.2020.103410

Reviewer 3 Report

Comments and Suggestions for Authors

In this article, the authors discuss the fluctuation of the spatial microcolumn distribution on the flow of DNA solution at high concentration, and prove that the hexagonal array has a smaller fluctuation scale than the quadratic array. The shedding of vortices is observed by comparing the fluid dynamics, which provides a good idea for engineering application. The article can be considered for publication in "Micromachines" after revising the following questions. The comments are below.

1. Under the title Geometry-dependent Elastic Flow Dynamics in Micropillar Arrays, the authors studied DNA sorting using three kinds of microarray structures. In fact, such a precisely designed microarray structure is widely used in the field of microfluidic. We suggest that the authors refine the current topic and link it to the research content - DNA sorting.

2. The Deborah numbers in Figure 1A and Figure 2A do not correspond exactly, and the authors need to be concerned whether this is a human error.

3. The microfluidic device is a powerful analysis carrier, and the authors can provide some appropriate renderings of the device, which can further enhance the richness of the article. Authors can refer to Biosensors and Bioelectronics, 2023, 230, 115586.

4. The flow rate unit used by the authors in the figures is partly µm/s and partly m/s. We suggest that the unit of flow rate be unified.

5. Many of the pictures in the article are provided with scales, and the authors also describe the dimensions. It is suggested that the authors directly mark the size on the scale, so that the reader could understand the information in the picture more intuitively.

Author Response

  1. Under the title Geometry-dependent Elastic Flow Dynamics in Micropillar Arrays, the authors studied DNA sorting using three kinds of microarray structures. In fact, such a precisely designed microarray structure is widely used in the field of microfluidic. We suggest that the authors refine the current topic and link it to the research content - DNA sorting.

RESPONSE:   The reviewer is correct that the original motivation for pursuing this line of research in our group is indeed the prospects of increasing throughput in microfluidic DNA sorting. However, we have broadened our views to include transport of viscoelastic fluids in general, where it is highly relevant to control the degree and structure of the fluctuations in the flow patterns. Therefore, we list both sorting and mixing as important applications. Finally, we are interested in a better understanding of the underlying physics of the waves, which we hope to reach by exploring various types of designs.

  1. The Deborah numbers in Figure 1A and Figure 2A do not correspond exactly, and the authors need to be concerned whether this is a human error.

We thank the reviewer for this pointing out. We went through the Deborah numbers and velocities given in the figures and were able to correct several additional inconsistencies in the figures. In the process we put together a summary of all data that will be attached to the data available in Harvard Dataverse to make it easier for any reader wishing to perform a careful analysis of our results.

  1. The microfluidic device is a powerful analysis carrier, and the authors can provide some appropriate renderings of the device, which can further enhance the richness of the article. Authors can refer to Biosensors and Bioelectronics, 2023, 230, 115586.

RESPONSE:   The reference to the article does not seem to be correct. The article with the stated article number can instead be found in the volume 239 of the journal [1]. However, we do not understand in which way it would have particularly strong relevance in light of the expected impact of our work, although we acknowledge the fact that on the long term, we target microfluidic transport of viscoelastic fluids in our work.

Yin, B., X. Wan, W. Yue, T. Zhou, L. Shi, S. Wang, and X. Lin, A portable automated chip for simultaneous rapid point-of-care testing of multiple β-agonists. Biosensors and Bioelectronics, 2023. 239: p. 115586

  1. The flow rate unit used by the authors in the figures is partly µm/s and partly m/s. We suggest that the unit of flow rate be unified.

RESPONSE:   The units have been changed in the text and in the figures and are now presented in a consistent way.

  1. Many of the pictures in the article are provided with scales, and the authors also describe the dimensions. It is suggested that the authors directly mark the size on the scale, so that the reader could understand the information in the picture more intuitively.

RESPONSE:   We do understand the point of stating the full information about the scale in the images. However, another important consideration is to minimize any distractions in the figures. Therefore, we have elected to give the size of the scale in each figure caption.

[1]       Yin, B., X. Wan, W. Yue, T. Zhou, L. Shi, S. Wang, and X. Lin, A portable automated chip for simultaneous rapid point-of-care testing of multiple β-agonists. Biosensors and Bioelectronics, 2023. 239: p. 115586. https://doi.org/10.1016/j.bios.2023.115586